



# Phosphorus regulates fungal biomass production in a Norway spruce forest

Juan Pablo Almeida[a], Lorenzo Menichetti[b], Alf Ekblad[c], Nicholas P. Rosenstock[d], & HåkanWallander[a]

[a] Lund University, Microbial Ecology, Dept of Biology, SE-223 62 Lund, Sweden

[b] Sveriges Lantbruksuniversitet (SLU), Department of Ecology, Ulls Väg 17, Uppsala, Sweden

[c] School of Science and Technology, Örebro University, SE- 701 82, Örebro, Sweden

[d] Center for Environmental and Climate Research, Lund University, SE-22362 Lund, Sweden

Corresponding author: Juan Pablo Almeida, jpalmeidava@gmail.com

Abstract

Ectomycorrhizal fungi (EMF) are important components of the soil microbial
communities and EMF biomass can potentially increase carbon (C) stocks by
accumulating in the soils as necromass and producing recalcitrant structures. EMF
growth depends on the C allocated belowground by the host trees and the nutrient
limitation on tree growth is expected to influence this allocation. Therefore, studying
EMF production and understanding the factors that regulates it in natural soils is
important to understand C cycling in forests.
Ingrowth meshbags are commonly used to estimate EMF production, but these
measurements might not reflect the total EMF production since turnover rates of the
hyphae are not considered. Here we estimated fungal production and turnover in
response to P fertilization in a Norway spruce forest where nitrogen (N) deposition
has resulted in phosphorus (P) limitation of plant production by using a combination
of meshbags with different incubation periods and with Bayesian inferences. To test
how localized patches of N and P influence EMF production and turnover we
amended some bags with a nitrogen source (methylene urea) or P source (apatite).



Additionally, the Bayesian model tested the effect of seasonality (time of meshbag
harvesting) on fungal production and turnover.
We found that turnover of EMF and was not affected by P fertilization or meshbag
amendment. P fertilization had a negative effect on EMF production in all the
meshbag amendments suggesting a reduced belowground C allocation to the
extramatrical mycelium under high P status. Apatite amendment significantly
increased EMF biomass production in comparison with the pure quartz bags in the
control plots but not in the P-fertilized plots. This indicates that P-rich patches
enhance EMF production in P limited forests, but not when P is not limiting. Urea
amendment had a general positive effect on EMF production, but this was
significantly reduced by P fertilization, suggesting that a decrease in EMF production
under high P status also will affect N foraging. Seasonality had a significant effect on
fungal production and the differences registered between the treatments were higher
during the warmer months and disappeared at the end of the growing season.
Many studies highlight the importance of N for regulating belowground C allocation
to EMF in northern coniferous forests, but here we show that the P status of the forest
can be equally important for belowground carbon allocation to EMF production in
areas with high N deposition.
Key words: Ectomycorrhizal fungi, fungal growth, fungal turnover, nitrogen
deposition, phosphorus limitation, apatite, methylene urea, Bayesian inference.






## 1 Introduction:

In terrestrial ecosystems forest soils are important reservoirs for carbon (Falkowski et al., 2000). Boreal forests contribute approximately 50% of the total forest carbon stock from which around 85% is stored in the soil (Malhi et al., 1999). At least half of the carbon stock in boreal soils originates from belowground carbon allocation through roots (Clemmensen et al., 2013) and a large portion of boreal forest primary production is allocated belowground by the trees (Gill & Finzi 2016). The carbon dynamics in forest soils are highly dependent on the soil microbial communities that either enhance C losses by degrading organic matter or increase C stocks by immobilizing C (Clemmensen et al., 2013). Filamentous fungi forming mycorrhizal associations for example, play an important role for C fluxes since some species have the capability to degrade a great variety of organic compounds while others can contribute to soil organic matter formation by releasing exudates that promote soil aggregation (Rillig, 2005) or produce slowly decomposing and highly melanized hydrophobic tissues (Almeida et al., 2022). The effect of EMF on soil microbial communities might not be trivial since up to 20% of the net primary production is allocated belowground to support the symbiosis (Hobbie, 2006). Therefore, ectomycorrhizal mycelium is expected to be a significant part of the soil fungal biomass and its production and turnover play an important role in forest carbon cycling and organic matter formation (Ekblad et al., 2013). For that reason, the development of methods that allows us to quantify EMF growth in forests natural soils is of paramount importance (Fernandez, 2021)

Therefore, understanding the factors that regulate the growth rates of filamentous fungi like EMF is important to understand carbon dynamics in soils. Growth rates of



free-living fungi from natural soils has been studied in laboratory by measuring
labeled acetate incorporated in the fungal membrane component ergosterol (Sheng et
al., 2022; Rousk and Bååth, 2007) or labeled water incorporated into DNA (Schwartz
et al., 2016). Quantifying growth (production) of EMF natural communities on the
other hand is more complicated since EMF are dependent on plant roots (Smith and
Read, 2008) and such measurements must be performed when the fungi is living in
symbiosis. Many studies have attempted to quantify EMF production *in situ* in forests
soils by using ingrowth meshbags and fungal biomarkers like ergosterol or PLFAs
(Wallander et al., 2013). In those studies, EMF production has been estimated based
on the standing fungal biomass measured in meshbags after a specific time of
incubation in the soil (Ekblad et al., 2013; Wallander et al., 2013; Wallander et al.,
2001). However, the standing biomass does not necessary reflect growth since the
standing biomass is the result of the interaction between fungal growth and the
residence time of the fungal mycelium in the meshbag (Ekblad et al., 2016). In order
to overcome these shortcomings, some studies have estimated EMF production and
mycelium turnover by repeated harvests of mycelial meshbags, applying ergosterol as
a marker of mycelial biomass and mathematical models to estimate the production
and turnover of EMM biomass (Hagenbo et al., 2021; Hagenbo et al., 2017) or,
combined with analyses of chitin, to enable estimates of production and turnovers of
both bio- and necromass (Ekblad et al., 2016). In these studies, the standing biomass
and necromass were analyzed in bags incubated over periods varying in length,
combining several shorter periods, one after the other, with overlapping longer
periods. Common assumptions in these studies were that EMF growth occurs at a
constant rate and that biomass and necromass were lost at constant exponential rates
(Ekblad et al., 2016).






By using this approach, Ekblad et al. (2016) tested the effect of nitrogen (N)
fertilization on EMF turnover and growth in a *Pinus taeda* forest. They reported that
fertilization significantly decreased both fungal standing biomass and growth but
turnover rates of biomass and necromass were not affected. It was suggested that the
decrease in fungal growth was regulated by changes in carbon allocation as a result of
an increase in soil fertility. These results are in line with evidence indicating that the
relative amount of carbon allocated to EMF is sensitive to plant nutrient status and
soil fertility (Gill & Finzi 2016). Thus, in boreal forests where N is the nutrient that
limits tree growth (Högberg et al., 2017), high amounts of carbon are invested below
ground to support ectomycorrhizal symbiosis to facilitate N uptake (Gill & Finzi

105 2016).


The role of N as limiting nutrient in high latitude forested ecosystems and its effect on
EMF is well known and has been described in several studies (Binkley & Högberg,
2016; Hedwall et al., 2013 ; Gill & Finzi, 2016) . However, it has been suggested that
anthropogenic N deposition can potentially change the forests nutrient requirements
and push the system toward phosphorus (P) limitation (Tarvainen et al., 2016; Du &
Fang, 2014; Akselsson et al., 2010; Vitousek et al., 2010). In fact, in a region with
high N deposition in southwest Sweden, Almeida et al. (2019) reported that P
fertilization had a stronger effect on tree growth than N fertilization, subverting the
expectation that N is the main nutrient regulating plant growth in northern forests. The
effect of the transition from N to P limitation on the below ground C allocation and
EMF growth has not been studied in natural soils, but P deficiency is expected to
increase EFM biomass to improve P foraging and uptake (Rosenstock et al., 2016;



Ekblad et al. 1995; Wallander & Nylund 1992). In a field study, Rosenstock et al.,
(2016) reported an increase in root- and ECM standing biomass in a Norway spruce
(Picea alba) forest limited by P compared to forests with sufficient P. In the field
study performed by Almeida et al. (2019) however, no effect on EMF standing
biomass was found in meshbags incubated for 133 days. Yet, since only the standing
biomass was measured and the turnover rates and production were not estimated, we
cannot exclude the possibility that P fertilization had an effect on EMF production, an
effect that cannot be detected by studying the standing biomass alone.

In this study, we aimed to improve our understanding of EMF production and
turnover in natural soils and to test how EMF production is affected when P is
limiting tree growth.   In the forest described by Almeida et al. (2019) we estimated
EMF production and turnover using the mathematical model of Ekblad et al. (2016)
with Bayesian inferences. Since EMF production is likely to follow root growth
which varies with season (Coutts & Nicoll, 1990 ; Walker et al., 1986), we performed
a more extensive incubation scheme and more frequent harvests of bags than in
Ekblad et al., (2016). This allowed us to test the model considering the treatments
effects (P fertilization and meshbags amendments) and also considering their
interactions with seasonality (time of the growing season). Because EMF growth is
subsidized by the host, in exchange for N and P, EMF production should be affected
by the nutrients found at the hyphal front. Indeed, EMF biomass in P-poor forests is
stimulated around localized patches of the P-rich mineral apatite (Rosenstock et al.,
2016; Berner et al., 2012; Hagerberg et al., 2003). Therefore, besides purely sand-
filled meshbags, we incubated meshbags amended with apatite or methylene urea





(referred as urea throughout the manuscript) in order to simulate soil N and P nutrient
patches respectively.

Our hypotheses were:

• P fertilization will decrease the biomass production of EMF mycelia.
• Apatite amendment will increase EMF biomass production in the control plots
but not in P fertilized plots.
• Urea amendment will increase EMF biomass production in the P fertilized but
not in the control plots.

**2 Materials and Methods**:

**2.1 Field site and fertilization treatments**
This study was performed at Tönnersjöheden forestry research station (56° 41' N, 13°
6' E, 80 m a.s.l.) with a mean annual temperature of 6.4 °C and a mean annual
precipitation of 1064 mm (Högberg *et al*., 2013). Soils are podzols developed in a
glaciofluvial parent material with a pH (in $H_2O$) of 4.05 and a C/N of 25.1 in the mor
layer (Hansson, 2011; Högberg *et al*., 2013). The forests consist of managed Norway
spruce (*Picea abies*) planted on former pastureland in 1979. The site is in southwest
Sweden with an N deposition of 14.5 kg $N^{-1}$ $ha^{-1}$ $yr^{-1}$ (Rosenqvist *et al.,* 2007), which
is high in comparison with most other forests in the country (Akselsson, 2010;
Högberg *et al*., 2013). The experiment consisted of 6 plots (30-40 m x 25 m); 3
control and 3 fertilized with 200 kg P $ha^{-1}$ of superphosphate (100 kg $ha^{-1}$ applied
twice in September 2011 and July 2012).





### 2.2 Experimental design


To estimate EMF mycelial production, ingrowth meshbags (Wallander *et al.,* 2001)
were incubated in the plots. The meshbags were cylindrical, 2 cm wide and 10 cm
long. They were made of 50 μm nylon mesh and filled with approximately 40 g of
quartz sand. Three different amendments in the meshbags were used: pure-quartz,
apatite-amended (quartz and 2% (w/w) crushed apatite mineral with a grain size of 50
to < 250 nm) and urea-amended (quartz and 0.5% (w/w) granulated methylene urea).
The mesh-bags were vertically installed into holes made with a soil corer (2 cm
diameter) with the upper end of the bag at level with the soil surface.

To calculate turnover rates and biomass production as done by Ekblad *et al.* (2016),
sequential meshbag incubations were performed. For a five-month period starting in
July 2015 and ending in November 2015, the meshbags were incubated for variable
periods of time (30, 60, 90, 120 or 150 days; Fig 1).

There were five different 30-day incubation periods. Four 60-day incubation periods
each overlapping with two 30-day incubation periods. Three 90-day incubation
periods each overlapping with three 30-day incubation periods. Two 120-day
incubation periods each overlapping with four 30-day incubation periods. One 150-
day incubation period overlapping with all 30-day incubation periods.
The bags incubated over 30 days were incubated sequentially and when one set of
bags was collected, a new set of bags was directly installed using the same holes as
the ones just emptied (Fig 1).
In each plot, a pure-quartz meshbag for each of the incubation periods described
above was placed along a 15 m long transect. The distance between each meshbag


was approximately 1.5 m. The apatite-amended and urea-amended bags were placed
10 cm (perpendicular to the long transect) at each side of the quartz meshbags. Three
15 m long transects were done to have three sub-replicates (for each set of bags) that
were pooled before further analysis to give one sample from each incubation period
and amendment (quartz, apatite and urea) per plot.

Each incubation period consisted of 54 meshbags (2 treatments C/P, 3 replicated
plots, three sub-replicates, three amendments (2 x 3 x 3 x 3 =54). In total, 810
meshbags were installed and collected according to their incubation period.

July         August       September     October      November









Figure 1: Overview of the incubation design. Different color bars represent the incubation time periods:
Yellow corresponds to 30 days, Light green to 60 days, Dark green to 90 days, Purple to 120 days and
Blue to 150 days of incubation. The arrows represent the points in time when the same holes from the
previous incubation were used to incubate the next set of meshbags.


Upon harvest, the meshbags were kept in an icebox until arrival to the laboratory
where they were stored at -20ºC.



The fungal cell membrane compound ergosterol, a proxy for fungal biomass, was
extracted and measured from 5 g of the pooled samples *as per* Bahr *et al.* (2013)
using high-pressure liquid chromatography (auto sampler L2130 with UV detector
L2400 by Hitachi, Japan). The fungal biomass was then expressed as µg of ergosterol
per gram of sand in the meshbag.

**2.3 Mathematical models**
The turnover rates and fungal biomass production were estimated applying the
mathematical model used in Ekblad et al. (2016). In this paper however the
mathematical model was tested under two assumptions:
Fungal production was dependent on the treatments alone (Model 1), or fungal
production was depended on treatments and sampling season (Model 2), allowing to
test for the interactions between treatment and seasonal effects.

Model 1:

This model works under the assumption that EMF production occurs at a constant rate
and that biomass is lost at a constant exponential rate (see Hagenbo et al., 2017 &
Ekblad et al., 2016). Briefly, the sum of the biomass during two sequential short
incubation periods is expected to exceed the biomass in an overlapping longer
incubation period due to an on average older mycelium and hence larger turnover in
bags with a longer incubation period.






The model in its differential form is defined as:

$$\frac{dB}{dt} = P - \mu \cdot B$$

*Equation 1*
Where $P$ is the production of new mycelium (in mass units), $B$ is the mycelium
biomass (also in mass units) and $\mu$ represent the mortality, the fraction dying over a
specified time-period (adimensional). This equation is solved over time as:

*Equation 2*
$$B(t) = \frac{P_k}{\mu_k} \cdot (1 - e^{\mu_k t})$$
In our case we assumed that both $P_k$ and $\mu_k$ are influenced by the fertilization
treatments, denoted here by $k$, and we therefore assigned a specific (unknown) P and
$\mu$ to each treatment in the Bayesian model.

Model 2:

Equation 2 has been utilized in other publications (Hagenbo et al. 2021; Hagenbo et
al. 2017; Ekblad et al., 2016) and one of the  main assumptions of this model is that
fungal production occurs at a constant rate. However, fungal production can vary
depending on the time of the year (Coutts & Nicoll, 1990 ; Walker et al., 1986) so we



tested a modification of the model by introducing an additional degree of freedom
into the model represented by the term $\beta_{k,j}$, dependent on sampling seasons ($j$) and
their interactions with treatments ($k$) so that the calibration can apply to each
treatment a correction for seasonality (independent from the other treatments). When
the term $\beta_{k,j} = 1$ then the model is equivalent to what described in eq. 1 and 2. We
utilized this model to decompose $P$ in two components, defining a new term $P'$:

*Equation 3*
$$P'_{k,j} = P0_k \cdot \beta_{k,j}$$

$P'_{k,j}$ corresponds to $P_k$ (if the distributions were perfectly symmetric the average for P
and P´ should converge to the same value) but the predicted biomass production now
is the results from the interactions between sampling season and treatments.

Eq. 3 is then substituted into Eq. 2 by substituting $P$ with $P'$. The resulting model is
equivalent to the one described by Eq. 2 for certain parameter combinations and
describes the same curve. The only difference is that now two components are used to
decompose the variance explained by the calibrated model in two separate terms: $P0_k$
which expresses the production variable with treatments only ($k$); and $\beta_{k,j}$ which
expresses the effects of seasonality and their interactions with treatments. $P0_k$  is now
equivalent to the production normalized by the seasonality effect $\frac{P'_{k,j}}{\beta_{k,j}}$ . By letting $P0_k$
and $\beta_{k,j}$ vary independently (therefore describing each point as a combination of $k$
and $j$) we avoid to make any strong assumption on the effect of seasonality (since we
are not imposing a parametric function of time to describe it but we let it free to vary





for each time point) or on its interactions with treatments (which are still free to vary
depending on the treatment), while on the other end we maximize the information we
can extract from the data by representing the interactions between the terms in one
single model calibration. If we instead relied on fully independent calibrations within
each subset of seasons × treatments we would have had to divide the data in $j \times k$
subsets where we would calibrate each model parameter independently, limiting each
calibration to a smaller number of samples.
**2.4 The calibration:**
The model was calibrated within a formal Bayesian framework, developed with the
Stan toolbox (Stan Development Team, 2021). This approach is based on a numerical
implementation of Bayesian statistics, which allows for a continuous update of the
knowledge while new data are developed, based on stochastic principles (through a
modification of the Metropolis-Hastings sampler). While we refer to relative
publications for technical details, the main assets of the method are that: a) we can
integrate and utilize previous information in the calibration, defining it as prior
probability distributions of model parameters (from now on, "priors), b) such
information is combined with the statistical information contained in the data to
determine the posterior distributions of model parameters and consequently
predictions, and such distribution is non-parametric (so not assuming any specific
shape but determined only by the available information). The methodology is
therefore extremely useful to combine multiple sources of information and very
valuable when information is scarce, and at the same time quite robust given that it
estimates detailed posterior probability distributions (which can be examined closely).





In our case the methodology allows us to draw information from publications. This
information is considered probabilistically. It does add information to our final results
(our posterior distributions), but such information is combined with the information
contained in our data. The chosen statistical approach updates the old information
with new data, and old and new information can be therefore compared.

We calibrated both a model with only Eq. 2 (so considering only treatment effects;
Model 1) and one considering Eq. 2 and Eq. 3 (considering treatments × seasonality
effects; Model 2).
Priors for $P_k$ and $\mu_k$ were derived from the literature, both expressed as normal
distributions with deviation prudentially estimated as 25% of the mean (please note
that this does not mean that the prior was limited within this range, due to the tails of
the normal distributions).
$P_k$  was expressed as
$$P_k \sim N(0.099, 0.099 \cdot 0.25)$$

While $\mu_k$ as
$$\mu_k \sim N(0.009, 0.009 \cdot 0.25)$$

Both priors were based on the mean fungal biomass production and turnover for forest
of similar age as the forest in the current study estimated by Hagenbo et al. (2017)
after unit conversion. The Bayesian system was run considering one independent $P_k$
and $\mu_k$ for each treatment.





When we also considered Eq. 3, priors for $P0_k$ were defined as the priors for $P_k$ while
priors for $\beta_j$ were set as uniform between 0 and 5.

339                                    $$\beta \sim U(0,5)$$

Please note that $\beta_j = 1$ means no seasonality effect, $\beta_j = 5$ means a five-fold increase
of production due to seasonality, while $\beta_j = 0$ means a complete halt of production
due to seasonal effect.

**2.5 Statistical analysis and probability distribution comparisons**
The standing biomass, data was tested for homogeneity of variances and normal
distribution using Levene's and Shapiro Wilk tests, respectively.  Analysis of the
variances (ANOVA), Tukey's Post-hoc test and Dunn analyses were performed on the
data to check for statistical differences between the fertilization treatments and
meshbag amendments. The Levene's and Shapiro Wilk tests, as well as ANOVA and
Dunn analyses were done by using R (R Core Team, 2014).

The stochastic approach of the Bayesian method produces Markov chains Monte
Carlo (MCMC) that represents a probability distribution with as many discrete
parameter values as iterations in the chains (in our case 10 independent chains of
10000 iterations, so a total of 100000 iterations), with a histogram that approximates a
continuous distribution (probability distribution). Thus, the predicted fungal
production and turnover for each treatment (fertilization regime and meshbag
amendment) is represented by a probability distribution.

The means of the probability distributions were calculated and the highest density
intervals of the estimated parameters were interpreted as confidence intervals at 95%



and 90% (Kruschke and Liddel, 2018). To test the significance of the treatments
(fertilization regime, meshbag amendment and season), the confidence intervals of the
probability distributions were compared.
**3 Results:**

**3.1 Mycelial standing biomass**
The standing biomass of mycelia in the meshbags was significantly affected by
incubation period (time of the year) (Kruskal-Wallis, $p < 0.0001$, $X^2 = 116.4$).
Biomass in one-month incubation mesh bags from July, August and September was
significantly higher than the biomass collected in October and November for both
control plots and P fertilized plots (Dunn´s test, $p < 0.001$, $X^2 = 26.1$) (Fig 2).
Biomass in two-months incubation mesh bags from July-August and August-
September was significantly higher than the biomass collected in September-October
and October-November for both control plots and P fertilized plots (Dunn´s test, $p <$
$0.001$, $X^2 = 27.7$; Fig 2). Fertilization significantly affected the standing biomass in
the quartz, apatite and urea-amended meshbags (Kruskal-Wallis, $p < 0.05$, $X^2 = 6.5$; $p$
$< 0.0001$, $X^2 = 18$; $p < 0.0001$, $X^2 = 15.5$; respectively). Phosphorus fertilization
reduced the standing biomass in all the incubation times (numbers of incubation days)
for apatite urea and amended meshbags (Fig 3). Apatite amendment significantly
increased the standing biomass in comparison with the pure-quartz bags in the control
plots after 60 and 150 days of incubation (Dunn´s test, $p < 0.05$, $X^2 = 18$; $p < 0.05$, $X^2$
$= 11.2$, respectively), and the effect of apatite was stronger after 150 days of
incubation where on average the biomass in the apatite bags was three-fold higher
than the biomass in the pure-quartz bags. Apatite amendment did not increase
biomass in the P-fertilized plots in any incubation time while urea amendment



increased biomass in most of the incubation times and for both C and P fertilized plots
(Dunn´s test, p < 0.05) (Fig 3).

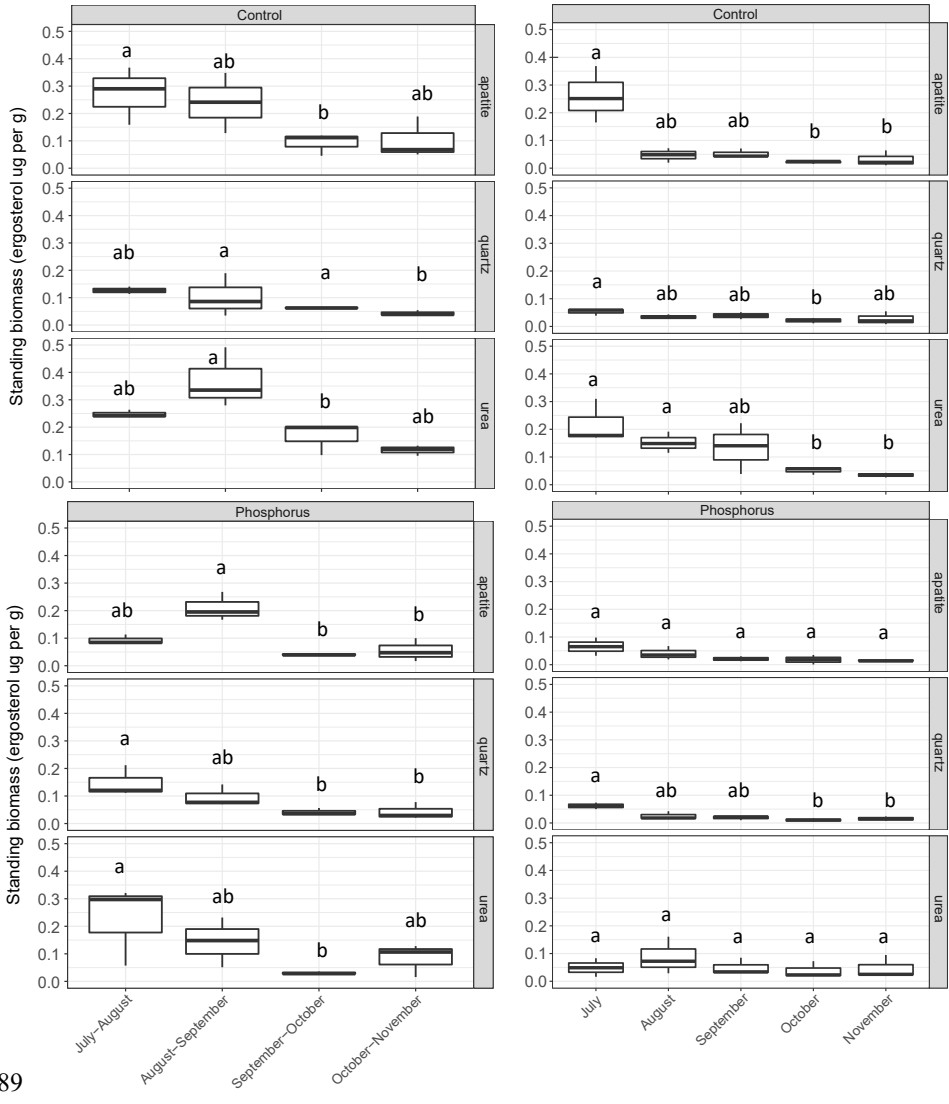


Figure 2: Boxplot of the standing fungal biomass in the meshbags incubated in the soil for 2 and 1
months. The boxes represent the interquartile range of the data (The central represents the median).
Higher and lower whiskers represent minimum and maximum range of the data (1.5 times the length of
the interquartile range). Lowercase letters represents statistically significant ($P<0.05$) differences
between the incubation periods according to Dunn´s test.






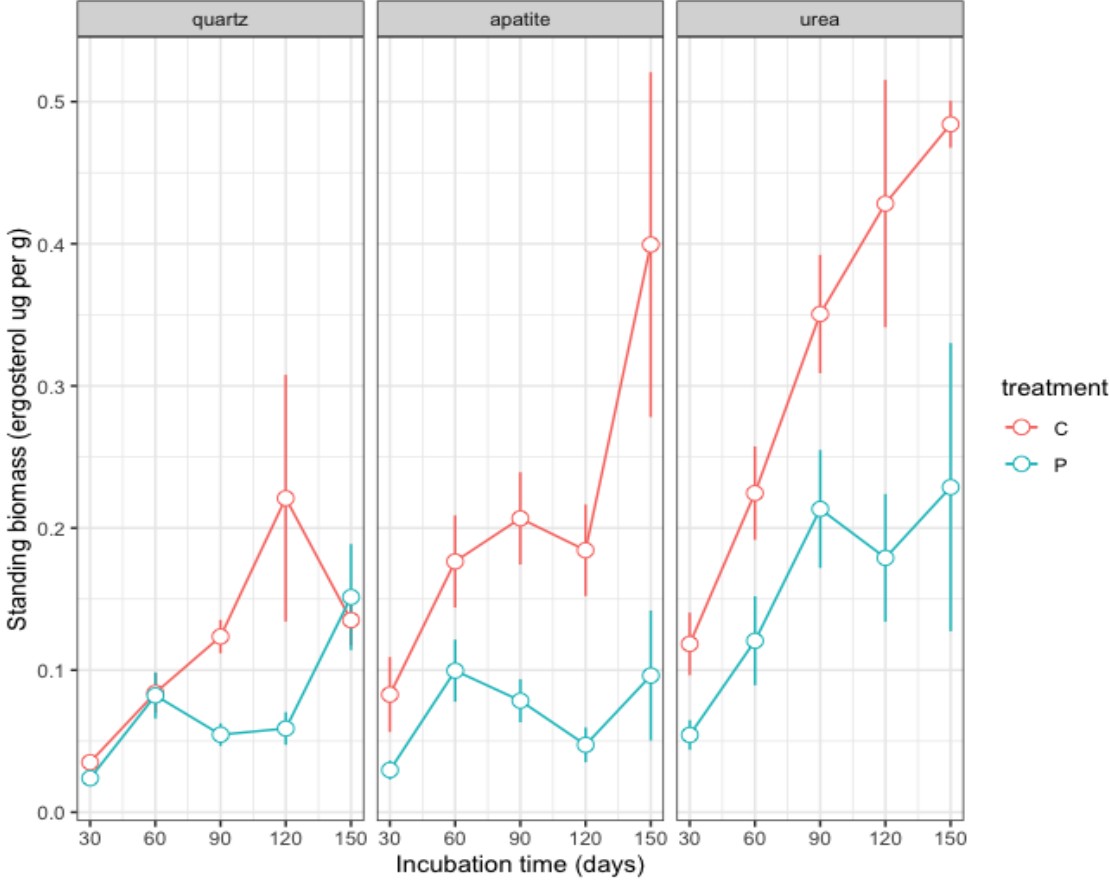

Figure 3: Standing fungal biomass in the three meshbags amendments (quartz-only, apatite and urea) and in the control plots (red symbols) and P-fertilized plots (blue symbols) and control plots during different incubation times (30, 60, 90, 120 and 150 days). The error bars represent the standard error of the mean.

**3.2 Fungal production and turnover rates (Model 1)**

The predicted fungal biomass production varied between the P-fertilized plots and the

control plots and between the meshbag amendments (Fig 4a). P fertilization

significantly decreased fungal production in all the meshbag amendments (urea and

apatite and quartz) (Table 1). In the P-fertilized plots the fungal production was

reduced to a third in the apatite and pure quartz bags in comparison with the prior

used to set the model (0.099 g m$^2$ day$^{-1}$). P fertilization caused a reduction on average

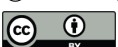



of 43% in the quartz bags, 60% in the apatite bags and 39% in the urea bags in
comparison with the control plots.

The meshbags amended with urea had the highest predicted biomass production in
both control and P-fertilized plots (Fig 4). Relative to the quartz bags, the urea
amendment doubled the production in both fertilizer treatments.  The apatite
amendment, in contrast, gave no significant change in production relative to the
quartz bags in the P-fertilized plots while a 35% increase was found relative to the
quartz bags in the Control plots (Table 1).

According to the mathematical modeling, the biomass turnover rates were not affected
by P fertilization or meshbag amendment (Fig 4 b).



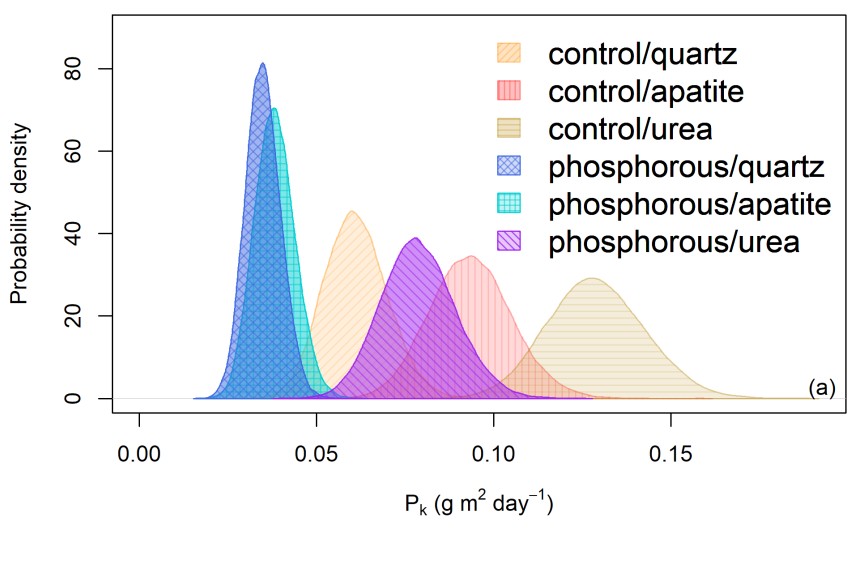

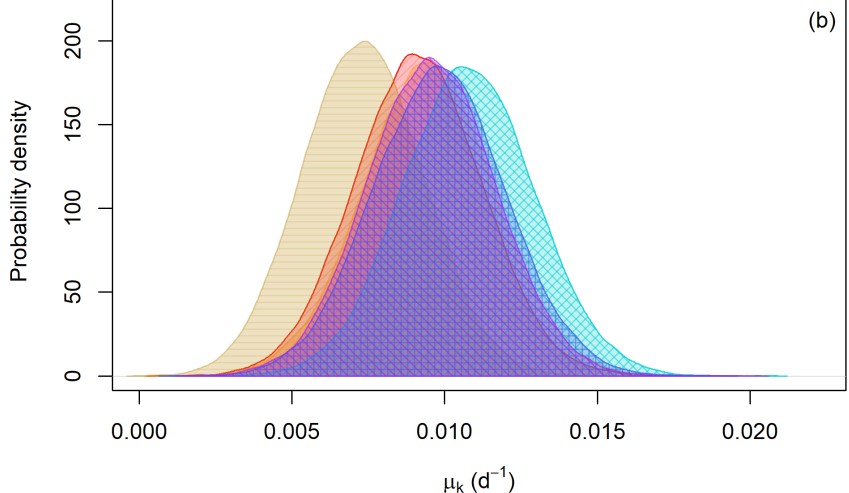


Figure 4: a) Probability distribution of the predicted fungal biomass production ($P_k$) (g m$^2$ day$^{-1}$) for the
different fertilizer treatments (Control and P fertilization) and meshbag amendments (quartz-only,
apatite and urea). b) Probability distribution of the turnover rates (day$^{-1}$) for the different fertilizer
treatments (Control and P fertilization) and meshbag amendments (quartz-only, apatite or urea).









Table 1. Mean of the fungal production in different treatments ($P_k$) estimated by Model 1. The Highest
Density Intervals (HDI, Kurshke and Liddel, 2018) represent the boundaries of each estimate at
different degrees of confidence.

| Fertilization and amendment | Mean fungal production ($g\ m^2\ day^{-1}$) | HDI low (95%) | HDI high (95%) | HDI low (90%) | HDI high (90%) |
|---|---|---|---|---|---|
| control/apatite | 0.094 | 0.072 | 0.117 | 0.075 | 0.113 |
| control/urea | 0.129 | 0.103 | 0.156 | 0.107 | 0.152 |
| control/quartz | 0.061 | 0.045 | 0.079 | 0.047 | 0.076 |
| phosphorous/apatite | 0.038 | 0.028 | 0.05 | 0.029 | 0.048 |
| phosphorous/urea | 0.079 | 0.059 | 0.1 | 0.062 | 0.096 |
| phosphorous/quartz | 0.035 | 0.026 | 0.045 | 0.027 | 0.043 |



### 3.3 Seasonal effect (Model 2)

The effect of seasonality as described by β had a positive effect on the predicted
fungal production and this effect was highest in July and decreased over time.
Moreover, the effect of β on fungal production differed depending on the fertilization
and on the meshbag amendment (Fig 5).

For example, in July the model suggests a seasonal effect increasing the predicted
fungal production by up to 5 times in the quartz meshbags from the P-fertilized plots
and up to 2.5 times in the urea meshbags in the control plots in comparison with the
apatite bags from the P-fertilized plots where season had no effect on fungal
production. The positive effect of sampling season on the fungal production, as
identified by the model, decreased in general with time and at the end of the growing
season (October and November) *β* had the same effect on all the samples
independently from the treatment (fertilization and meshbag amendment).



451 Even though the $\beta$ probability distributions of the different treatments were not

452 significantly different, the effect of the season on biomass production was important

453 and when we decompose fungal production by seasonality ($P'_k$), the differences in

454 fungal production between P fertilized and control plots and between the meshbag

455 amendments are present only early in the season (July, August) and disappear in

456 September October and November (Fig 6).

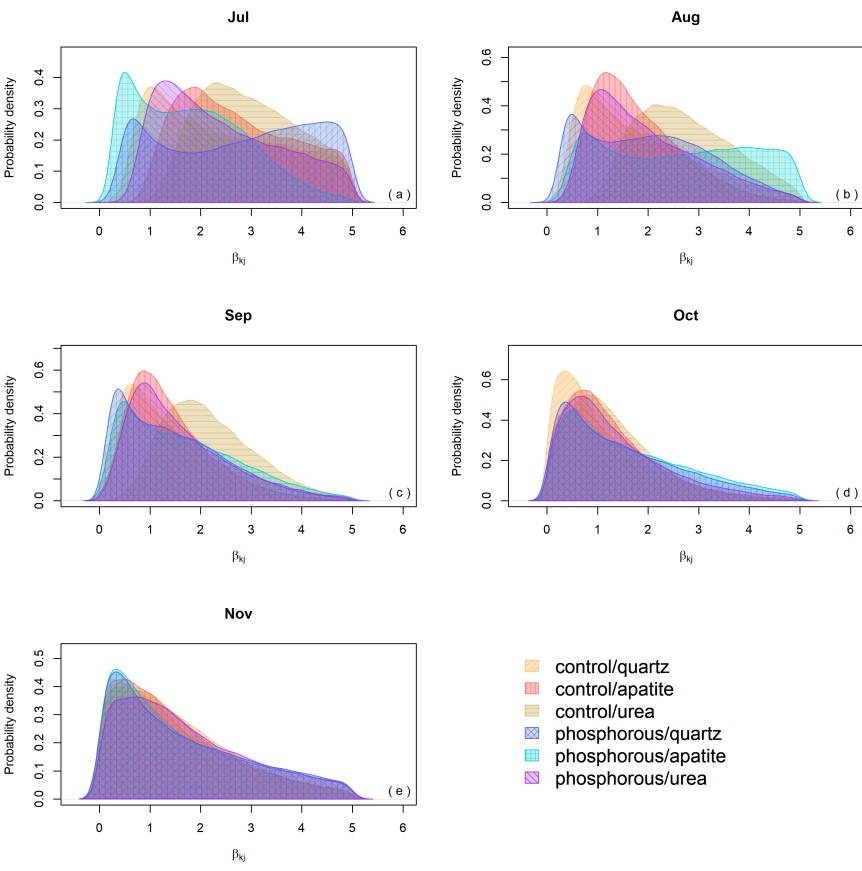


458 Figure 5: Seasonality effect on biomass production expressed by the $\beta$ parameter for the different
459 months of the growing season.





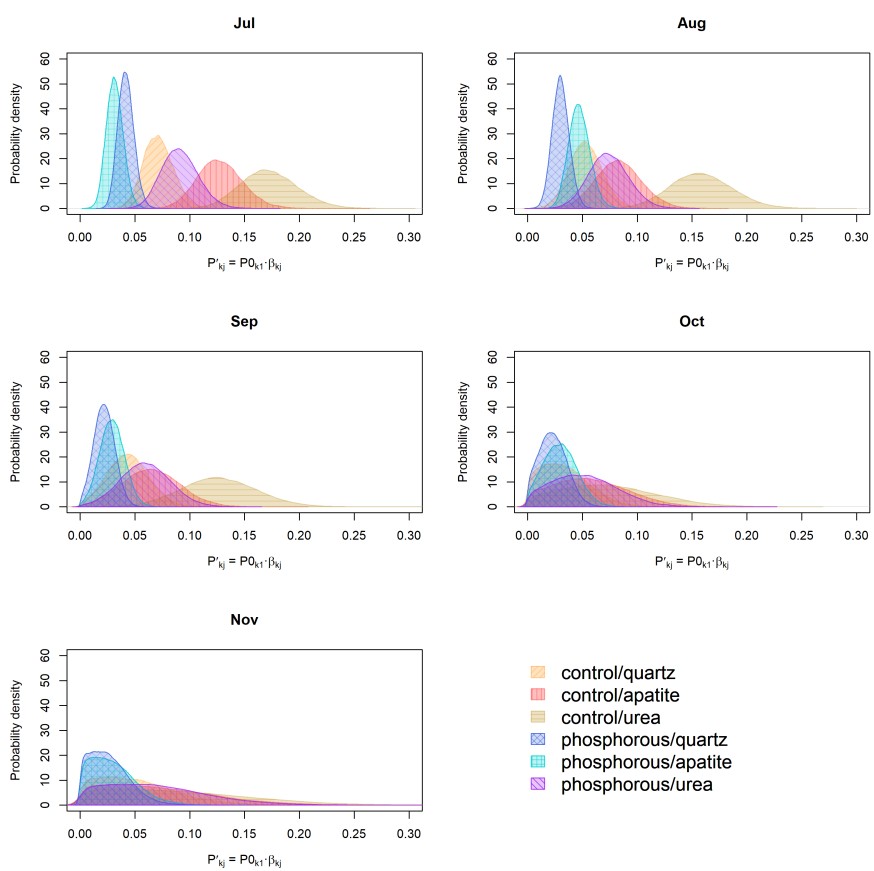


Figure 6: Probability distribution of P´$_k$ (g m$^2$ day$^{-1}$) for the different months of the growing season.













**4 Discussion:**

**4.1 Effect of P fertilization on fungal biomass production and turnover**
In support of our first hypothesis, fungal biomass production declined in response to P
fertilization in all meshbag amendments (Fig 4a). These results contrast with those of
Almeida *et al.* (2018) who tested the effect of P fertilization on the fungal standing
biomass in the same plots as in the present study.  This contrast is not depending on
variation in turnover rates between control and P fertilized plots since mortality was
not significantly affected by fertilization as shown in the current results. In the present
study, P had a negative effect on the fungal standing biomass in most of the
incubation periods (Fig 3). The fact that more incubation periods and a larger number
of bags were used makes the present study more reliable. Thus, the standing biomass
of one given incubation time might not truly reflect the effect of fertilization on fungal
growth. The use of the sequential incubation method and the mathematical model
allowed us to have a more robust estimate of the effect of P fertilization on the
extramatrical mycelium in this forest. P as a nutrient regulating fungal growth in
boreal forest was not reported before.

Fertilization experiments have been largely used to evaluate the effect of soil fertility
and nutrient status of the trees on carbon allocation and EMF production (Bahr et al.,
2015; Ekblad et al., 2013). However, studies on the effect of nutrient additions on
EMF in boreal forests have predominantly focused on N fertilization (Leppälammi-
Kujansu et al., 2013) probably because N is the most common limiting nutrient in
boreal forests (Högberg et al., 2017). Therefore, the effects of P additions alone on
boreal forests have not been widely tested. Due to the steep increase in anthropogenic





C and N inputs relative to P inputs, plant nutrient stoichiometry can be altered and
lead to unbalanced nutrition and lead to P limitation (Jonard et al., 2015; Peñuelas et
al., 2013). Indeed, P fertilization enhanced tree growth in the forest where this study
was performed as reported by Almeida et al. (2019).
Belowground carbon allocation is expected to be reduced by P fertilization when the
system is P limited (Gower & Vitousek 1989; Keith et al. 1997) leading to a decrease
in EMM production (Treseder, 2004). We propose that the decreased fungal
production in the P-fertilized plots in our study is a result of a decrease in
belowground C allocation due to alleviated P limitation that reduced tree dependency
on EMF for P foraging and acquisition.

This reduction in fungal production was not trivial and P fertilization decreased the
predicted fungal production to a third in comparison with the fungal production of a
forest of similar age estimated by Hagenbo et al. (2017) (0.099 g m$^2$ day$^{-1}$). More
studies on the effect of P fertilization alone in northern forested ecosystems receiving
high levels of N deposition should be performed to test if P-limitation is widespread
in these ecosystems as reported in this single forest.

A decrease in EMF production caused by fertilization might reflect a change in the
fungal communities. When there is a decrease in belowground C allocation, some
EMF species that require less C for growth and produce lower biomass relative to
other members of the community might be selected. In the previous study in the same
research forest (Almeida et al., 2019), EMF fungal communities from soil and
meshbag samples significantly changed after P fertilization and P + N fertilization
respectively. In particular, the most abundant EMF species *Tylospora asterophora*

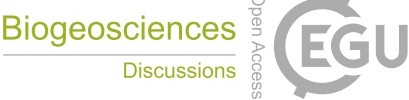

increased when the plots were fertilized with P or P + N. *Tylospora asterophora,* a
short exploration type (Agerer & Raidl, 2004), is expected to produce less biomass
than species with long exploration mycelia. Therefore, it is possible than an increase
of this species relative abundance in the meshbags of the present study might be
related to the lower growth detected in the P fertilized plots. It is also expected that
turnover rates vary depending on the species traits of the EMF community (Ekblad *et*
*al.,* 2016). For example, certain traits like rhizomorphs are expected to have longer
life span in comparison with smooth and short exploration type mycelium (Pritchard
*et al.,* 2008; Ekblad *et al.,* 2016). The significant increase of *T. asterophora* after P
fertilization could increase the overall mycelial turnover rate in these.  However, there
was not a detectable effect on the turnover rates between control and P fertilized plots.
In a tree age chronosequence study in a boreal forest in central Sweden, Hagenbo et
al. (2018) reported no clear pattern in exploration types despite a significant shift in
fungal community composition and turnover with forest age.  This suggests that
factors other than exploration types are also important to explain turnover rates.
Species-specific traits like mycelial life span, the degree of internal autolysis and the
amount of melanin in cell walls could potentially affect biomass turnover in EMF
communities (Hagenbo et al., 2018; Fernandez et al., 2013).

**4.2 Effect of nutrient amendment on biomass production and turnover**
Both nutrient amendments (urea and apatite) increased EMF production in
comparison with the quartz-only meshbags in the control plots. This is consistent with
mesocosm experiments that have shown that when organic (Wallander & Pallon,
2005; Leake et al., 2001; Bending & Read 1995  ) and mineral nutrient patches (Smits
et al., 2012 & Leake et al., 2008) are colonized by EMF, mycelial branching and





proliferation increase to explore the nutrient patch. In support of our hypothesis,
apatite amendment increased EMF production in comparison with the pure quartz
bags but only in the control plots. Our results are consistent with the view that trees in
the control plots are P limited, and that they allocate more resources to the EMF when
exploring a P source like apatite. When P limitation is alleviated by fertilization
however, there is probably a decrease in C allocation to the root symbionts which
could cause the reduced EMF colonization in the apatite bags.  This is supported by
other studies reporting that apatite amendment increases EMF standing biomass in
meshbags under P-poor conditions (Rosenstock et al., 2016; Berner et al., 2012; Hedh
et al., 2008; Hagerberg et al., 2003). In a fertilization study in nearby plots in the same
forest, Bahr et al., (2015) showed that apatite addition stimulated EMF standing
biomass in mesh bags, in control and in N-fertilized plots, but when N was added in
combination with P, on the other hand, no significant differences were found between
apatite amended and pure-quartz bags. All together these results provide evidence that
EMF growth is responsive to P nutrient patches, but this response is depended on the
P demand of the host.

From the two nutrient amendments, urea had the highest effect on fungal growth and
both in the control and P-fertilized plots. From a phytocentric point of view it could
be expected that EMF growing on a P rich source like apatite are rewarded with more
C from the P limited trees than EMF colonizing N bags. The stronger response of
EMF growth to the N nutrient patches than to P nutrient patches in the P-limited
control plots suggests that even though the forest is limited by P, N still has an
important effect on the growth of EMM.



It is possible that P limitation results in a general increase in C allocation to the root
symbionts and the C invested by the tree is delivered indiscriminately among its
fungal symbionts, independently of the nutrient patch they are colonizing.
Probably this is not surprising since N is needed by fungus and plant alike and in
order to produce biomass to forage for P and enzymes to mineralize it, EMF requires
N. Thus, N uptake can improve the P nutrition of the mycorrhizal system and positive
feedback between plant and fungus might happen.

Despite the strong effect of N patches on fungal growth, P fertilization decreased
growth in all meshbags independent of the amendment. EMF communities in forests
are diverse and composed of species with different abilities to mineralize the different
nutrients present in the soils (Lilleskov et al., 2011). By amending the meshbags with
different nutrient types, fungal communities are selected depending on the nutrient
added (Almeida *et al.,* 2019; Rosenstock *et al.,* 2016). The consistent effect of P
fertilization on both nutrient patches and even in the barren quartz-only bags suggests
that P fertilization affects growth of different EMF communities alike and reduces
nutrient foraging for both N and P. This is consistent with the idea that alleviated P
limitation results in a general decrease of C delivered to the roots and the mycorrhizal
symbionts.

Previous studies on EMF growth have focused on fungal biomass collected from
meshbags filled with acid washed sand (see Hagenbo et al. 2021; Hagenbo et al. 2017;
Ekblad et al 2016). However, since the pure quartz mesh bags are devoid of nutrients
(except probably for dissolved organic material entering the bags during incubation),
they might underestimate EMF production in soils. Moreover, in soils most of N and



P are heterogeneously distributed in nutrient patches (Hodge, 2006). For this reason,
amending the meshbags made possible to imitate the soil nutrient conditions that
influence EMF growth in forests and to understand how the nutrient regimes (both as
inorganic nutrient fertilization and as nutrient patches) affect EMF production. In fact,
the EMF growth in this study was influenced both by the nutrient at the hyphal front
(N and P amendment) and by the C provided by the roots (as shown by the effect of P
fertilization).

There were not differences in mycelium turnover between the different meshbag
amendments. This contrast with previous studies showing that the nature of a nutrient
patch could also affect hyphal turnover (Ekblad et al., 2013; Jansa et al., 2011).
Mineral substrates like feldspar have been shown to maintain fungal growth for up to
15 weeks (Rosling et al., 2004), while organic nutrient patches have been shown to
sustain fungal growth for around 5 weeks (Bending & Read 1995). Therefore, organic
substrates like urea are expected to be quickly depleted in soils. As a result, the EMF
hyphae is expected to autolyse and transfer the nutrients to other locations of the
exploring mycelium faster than during the slow weathering of mineral substrates like
apatite (Ekblad *et al.,* 2013 ; Jansa *et al.,* 2011). Therefore, it should be expected that
the apatite bags show lower turnover rates than the urea bags. In the present study
however, we could not detect differences between the two nutrient patches.  The
material used to amend the urea meshbags in this study is methyleneurea which is a
slow N release molecule. Thus, methylene urea is hydrolyzed to ammonium at a
slower rate than the urea molecules (Högberg et al., 2020). Therefore, even if there is
evidence that some EMF species can directly consume urea (Morel et al., 2008;





Yamanaka, 1999), these slow releasing nutrient sources might require a more
persistent mycelium than other organic sources.

Additionally, previous mesocosm experiments have shown that when EMF mycelium
grows on sand, longevity is enhanced in comparison with EMF growing on nutrient
patches (Wallander & Pallon 2005). Nutrient patches enhance growth and metabolic
activity of EMF, which may enhance turnover rates. For example, Bidartondo et al.
(2001) tested ectomycorrhizal growth response to apatite and ammonium in growth
chambers with EMF colonized *Pinus muricata* seedlings.  It was found that apatite
and ammonium addition increased the respiration rates of EMF, which could be taken
as an indication of higher metabolic activity and probably higher mortality. Thus, it
can be expected that EMF growing on the quartz bags have lower turnover than the
mycelium colonizing the nutrient amendments, but this was not the case in this study.
These discrepancies relating EMF turnover rates between the current and previous
studies might be caused by shortcomings on the sequential incubation method used
for the model in this paper. This method relies on the premise that the sum of the
biomass from meshbags incubated for short continuous periods should exceed the
biomass from meshbags incubated from a long incubation time. However, in a
number of cases the mycelial biomass from a long incubation period was greater than
the sum of the consecutive shorter intervals. This could be caused by a delay or a lag
phase in fungal colonization inside the bags. It is possible that when a meshbag was
collected and the same hole was used to replace a new bag (Fig 2) there was a lag
phase before the hyphae could colonize the newly placed meshbag (Wallander et al.,
2013). Thus, those data points could have created noise in the data making the
turnover estimates less robust. In any case, if turnover in the EMF communities



colonizing the nutrient amended bags is higher (as suggested by previous studies), and
was underestimated in the current study, then the high standing biomass measured in
the urea and apatite bags can only be explained by even higher EMF production than
the predicted in these results.

**4.3 Seasonal effects on fungal growth**
The general assumption of Model 1 is that fungal growth occurs at a constant rate.
However, this approximation has some limitations, since seasonality usually affects
the amount of C allocated to the roots (Coutts & Nicoll, 1990) and consequently EMF
root colonization (Walker et al., 1986). Indeed, the standing fungal biomass in the
mesh bags peaked in July and decreased over autumn (Fig 2). In this paper Model 2
allowed the predicted fungal growth to vary both with seasonality and with the
treatments (P fertilization and meshbag amendment). The introduction of these
different dependencies in the model allowed us to test for the interactions between
treatment and seasonal effects. It must be noted that the predicted fungal growth
resulting from Model 1 is not incorrect and truly reflects the fungal growth
differences between the treatments. However, by including seasonality in Model 2, we
could detect that those differences predicted earlier were highly dependent on the
season. Indeed, fungal growth not only increased early in the season, but the
magnitude of this increase depended on the treatments (Fig 5). Therefore, the
differences in biomass production between the fertilization regime and meshbag
amendments were significant only early in the season (Fig 6).

The fungal biomass seasonal peak reported in the current paper contrasts with
previous studies that have reported that the standing biomass in meshbags collected





from a *Pinus sylvestris* (Hagenbo et al., 2021; Wallander et al., 2001), *Pinus pinaster*
(Hagenbo et al., 2021) and *Picea albies* (Wallander et al., 2001) forests was higher
during the autumn season. However, in a study performed in the same experimental
area as the present study, Wallander *et al.* (2013) found that the standing biomass in
September-October incubations was lower than the standing biomass in July-August
incubations. It has been reported that different EMF species have different seasonal
peaks (Castaño *et al.,* 2017; Iotti *et al.,* 2014; De la Varga *et al.,* 2013) which could
explain the differences in fungal growth between previous studies and the current
experiment. Our results are also consistent with those from Coutts & Nicoll (1990)
who found that the mycelium extension of *Laccaria proxima* and *Telephora terrestris*
inoculated in *Picea sitchensis* peaked during July and decreased in autumn. The
mycelial extension was associated with soil temperature, which peaked early in the
growing season.

It could be also possible that non-mycorrhizal fungi had an important contribution to
the fungal growth detected in the current study. The meshbag system favors the
growth of EMF over non-mycorrhizal fungi as it has been shown in some studies
(Almeida et al., 2018; Rosenstock et al., 2016; Berner et al., 2012) which might
suggest that fungal growth in this study is influenced mostly by EMF. However, it has
been shown that the shorter the time period a meshbag remains underground the
higher the proportion of non-mycorrhizal fungi inside the bags (as measured by the
proportion of non-mycorrhizal DNA in Hagenbo et al., 2018). Non-mycorrhizal fungi
growth has been reported to respond positively to temperature (Pietikäinen et al.,
2005) which might imply that during the warmer months of July and August
filamentous non-mycorrhizal fungi growth was promoted and there was a higher



colonization of this fungal guild inside the meshbags. Even so, the effects of the P
fertilization and meshbag amendment on fungal growth were higher early in the
season which might imply that the seasonal effect seen in the current study is
explained mostly by EMF as it was discussed previously.

In conclusion, EMF production was strongly reduced when P was added to the
forests, suggesting a decline in belowground C allocated by the trees when the P
limitation was alleviated. This decline affected not only the foraging for P (apatite)
but also foraging for N (urea). The strong negative effect of P fertilization on EMF
production suggests a central role of P in regulating EMF biomass production in N
rich forests. Moreover, the effect of the reduced belowground C allocation and the
nutrient patches on EMF growth was significant only in the warmest months of the
growing season suggesting an important effect of seasonality on EMF growth
dynamics and nutrient uptake.

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
