# Peer review of "Phosphorus regulates fungal biomass production in a Norway spruce forest"

_Biogeosciences, 2022_

## Author Response (AR1)

**Reviewer 1:**

We thank the reviewer for the comments that have improved the understanding of the manuscript. In particular, the choice of the right terminology made the message much clearer.

**Comment:**

However, there are discrepancies between the expectations the title stirs by its wording ("fungal biomass"), the aim formulated in the introduction (L128-L130), the hypotheses (L146-152) and the methods. Wording and presentation of goals of the manuscript should be unified and unique aims/results be highlighted: 1) please choose a term that can be derived from the methods and use it uniformly, e.g., "EMF biomass production", because the distinction from similar terms ("fungal biomass") throughout the manuscript is unclear.

**Reply:**

It is indeed a fair point. The main goal of the paper was to analyse EMF biomass production, so we used the meshbag methods which is well known for selecting EMF from other EMF guilds. However, we lack non-mycorrhizal controls (as it was pointed out in one of your comments below) so we could not confirm the assumption that our measured and predicted biomass was entirely EMF. Therefore, the term *fungal* instead of EMF was used.

However, as it is discussed in the manuscript it is very likely that **most of** the fungal growth registered in the current study is of EMF origin. Therefore, we have decided to use the term **EMF production** instead of fungal production. We have changed the terminology accordingly in the manuscript.

Additionally, we added some motivation for the use of this terminology in the materials and methods:

The fungal cell membrane compound ergosterol, a proxy for fungal biomass, was extracted and measured from 5 g of the pooled samples as per Bahr et al. (2013) using high-pressure liquid chromatography (auto sampler L2130 with UV detector L2400 by Hitachi, Japan). It was assumed that after incubation in the soil the meshbags contents were dominated by EMF as it has been shown by metabarcoding (Almeida et al., 2018; Rosenstock et al., 2016; Berner et al., 2012; Wallander et al. 2010; Hedh et al. 2008) and isotopic studies (Wallander et al., 2001). Therefore, the fungal biomass collected was expected to be of EMF origin.

In the discussion we have added a new section disusing this further.

See one of the comments below that talks about the lack of non-mycorrhizal controls..

**Comment:**

2) the hypotheses are unconnected bullet-points, which is making them very generic; please integrate them into the last part of the introduction together with the reasoning behind them, especially as a body of prior studies followed similar questions.

**Reply:**

The hypotheses have been incorporated into the last part of the introduction's body:

In this study, we aimed to improve our understanding of EMF production and turnover in natural soils by testing how fungal production collected from ingrowth meshbags is affected when P is limiting tree growth. In the forest described by Almeida et al. (2019) we estimated EMF production and turnover using the mathematical model of Ekblad et al. (2016) with Bayesian inferences. **Our first hypothesis was that P fertilization will decrease EMF biomass production in this P limited forest as a result of the limitation being alleviated**.

In addition, because EMF growth is subsidized by the host, in exchange for N and P, EMF production in the meshbags production should be affected by the nutrients found at the hyphal front. Indeed, EMF biomass in P-poor forests is stimulated around localized patches of the P-rich mineral apatite (Rosenstock et al., 2016; Berner et al., 2012; Hagerberg et al., 2003). Therefore, besides purely sand-filled meshbags, we incubated meshbags amended with apatite or methylene urea (referred as urea throughout the manuscript) in order to simulate soil N and P nutrient patches respectively. We expected that the nutrient patches will increase EMF biomass production depending on fertilization. In particular: apatite amendment will increase EMF biomass production in the control plots but not in P

**fertilized plots (second hypothesis); and urea amendment will increase EMF biomass production in the P fertilized but not in the control plots (third hypothesis).**

Finally, since belowground C allocation follows the three phenological cycles (Endrulat et al., 2016), EMF production is likely to vary with season peaking in autumn (Hagerberg & Wallander, 2002 ;Wallander at al., 2001; Hagenbo et al., 2021), we performed a more extensive incubation scheme and more frequent harvests of bags than in Ekblad et al., (2016). This allowed us to test the model considering the treatments effects (P fertilization and meshbags amendments) and also considering their interactions with seasonality (time of the growing season). Therefore, our fourth hypothesis was that EMF biomass production will be higher in autumn than in summer.

Comment:

3) turnover/seasonality was a central aspect in the study, it should be involved in a hypothesis, too.

**Reply:**

Seasonality has been included as a hypothesis in the last part of the introduction's body. See the comment above.

Turnover is an important part of the study however we aimed to measure EMF growth (production) when a forest is P-limited. In that sense turnover rates were not part of the hypothesis but it was a mean to improve our measurements of EMF biomass.

Specific Comments:

- Title: "fungal biomass production" includes all types of fungal lifestyles, including the quantitatively very important group of saprotrophs. However, most of the manuscript uses the

term "EMF production". Therefore, I suggest to change the wording or explain in detail in the text how EMF production could be a proxy for other fungal lifestyles.

Reply:

We changed to:

Phosphorus regulates ectomycorrhizal fungi biomass production in a Norway spruce forest

- Abstract:

L12: To me, it is not clear, how you arrive at "fungal production" here, when you specify in L10 that you estimated EMF production. This should be clarified throughout the manuscript.

Reply:

Changed to:

Fungal mycelium collected from ingrowth meshbags is commonly used to estimate EMF biomass, but these measurements might not reflect the total fungal production since turnover rates of the hyphae are not considered.

- Introduction:

L86: EMM abbreviation is not explained. Do you mean extrametrical mycelium? (same in L502, L569)

Reply:

Changed

L97: Ekblad et al .2016 do not use the term fungal standing biomass

Reply:

I have change it to EMF standing biomass that is the term they used.

**- Methods:**

Experiments: was there any kind of control to examine the share of non-EMF fungi in the ingrowth bags, like ingrowth bags in a root-free area of soil, or amplicon sequencing of the EMF that were found in the ingrowth bags? – It would be very helpful to have clear knowledge to which degree this experiment was able to capture the term "EMF biomass production" used throughout the manuscript.

**Reply:**

Unfortunately, no, we do not have non-mycorrhizal controls in our study.

We have reorganized the last part of the discussion and created a new section where I acknowledge and discuss this weakness of the study:

**4.4 Potential non-mycorrhizal growth in the meshbags**

It could be also possible that non-mycorrhizal fungi contributed to the fungal growth detected in the current study. The main assumption that the ergosterol in this experiment comes mostly from EMF relies on previous evidence that the meshbag system favors the growth of EMF over non-mycorrhizal fungi (Almeida et al., 2018; Rosenstock et al., 2016; Berner et al., 2012; Wallander et al. 2010; Hedh et al. 2008; Wallander et al., 2001). However, it has been shown that the shorter the time period a meshbag remains underground the higher the proportion of non-mycorrhizal fungi inside the bags (as measured by the proportion of nonmycorrhizal DNA in Hagenbo et al., 2018).

Thus, non-mycorrhizal fungi growth could partially explain the seasonal effect detected as this fungal guild has been reported to respond positively to temperature (Pietikäinen et al., 2005). Unfortunately, the current study lacks non-mycorrhizal biomass controls (ie: fungal biomass from ingrowth bags collected in a trenched root-free area) that can be used to estimate the contribution of non-mycorrhizal fungi. Therefore, we cannot rule out the possibility that part of the ergosterol measured in the bags came from non-mycorrhizal fungi. Even so, the significant negative effect of P fertilization on all the meshbag types suggests that the decrease in fungal growth might be related to reduction in C allocation by the trees as discussed earlier. Moreover, the effects of the P fertilization and meshbag amendment on fungal growth were higher early in the season which might imply that the seasonal effect seen in the current study is explained mostly by EMF.

It must be noted nevertheless that a potential reduction in belowground C allocation could decrease root activity and possibly root exudates which might reduce labile sugars in the soils affecting saprotrophic fungi as well. Further studies are necessary to evaluate the effect of P limitation on root dynamics and other members of soil microbial communities.

L188-190: drilling a new hole and placing an ingrowth bag in it and re-placing an ingrowth bag by putting it into an existing hole seem to be two different kinds of disturbance. Is there any knowledge on this?

Reply:

In both cases (placing and replacing) there will be severing of hyphal connections either by the soil corer or by removing a bag from the soil. To our knowledge there is no information which one produce the highest disturbance. However, introducing the soil corer that is sharp should make a clean cut in the mycelium. After that a volume of soil which contains some mycelium is removed by the soil corer.

When a bag is removed in order to place a new one, the hyphal connections are also broken and some mycelium is removed inside the meshbag that is taken up.

Since, the meshbag diameter is the same as the soil corer, the broken mycelium has to recolonize the same space in both cases (placing and replacing). Therefore, the disturbance should be very similar.

Models: I am not versed in modelling and Bayesian inference, wherefore I could not review parts based on this in detail. However, the models seem well thought trough. Unfortunately, there is no explanation of data sources. The statements "the methodology allows us to draw information from publications" (L313) and "Priors for ð '*f*k and 𠜇k were derived from the literature"

(L322) are too vague. How was this done and which publications where used? Please explicitly state if this is based on data from Hagenbo et al. 2017 (L334) or new estimations. -- All data sources should be clarified and the data made available.

Reply:

Fair enough. We have made it clearer:

In our case the methodology allows us to draw information from previous studies. In particular, we used information from a EMF production study in a conifer forest by Hagenbo et al. (2017).

**AND:**

Priors for  $P_k$  and  $\mu_k$  were derived from the mean EMF biomass production and turnover for a forest of similar age as the forest in the current study estimated by Hagenbo et al. (2017) after unit conversion. Both priors were expressed as normal distributions with deviation prudentially estimated as 25% of the mean (please note that this does not mean that the prior was limited within this range, due to the tails of the normal distributions).

- Results:

L380: do you mean "... apatite, urea, and not amended meshbags"?

Reply:

No, there was a mistake I meant:

for the apatite and the urea amended meshbags (Fig 3).

L389-294 / Fig. 2: Please mention the number of data-points for each boxplot, either by plotting or mentioning n in the figure caption. As I understood the methods section, each box in figure 2 is resembling three samples (one pooled sample per plot), accordingly n = 3. In this case, a boxplot is not very useful in summarizing the data and another type of graph could better be chosen.

**Reply:**

The boxplots were changed to circles with standard error bars.

The number of samples (n=3) is now specified in the figure legend.

L398-403 / Fig. 3: having connecting lines between the points, on first glance indicates a time series with one starting point, but in fact there are for example several 30 and 60 day starting points and the samples are independent of each other. Therefore, deleting the lines (or making them dotted) would be useful. Additionally, labelling the x-axis with "incubation time of ingrowth bags [days]" would also help for understanding.

**Reply:**

The lines were made dashed, and the label changed as suggested.

- Discussion:

L481-482: "The fact that more incubation periods and a larger number of bags were used makes the present study more reliable." Please clarify: more reliable than what study? And what is the difference implied by "more ... periods" and "larger number"?

Reply:

Indeed, it was not so very clear. That statement was deleted. Only this remains:

Thus, the standing biomass of one given incubation time might not truly reflect the effect of fertilization on EMF growth. The use of the sequential incubation method and the mathematical model allowed us to have a more robust estimate of the effect of P fertilization on the extramatrical mycelium in this forest.

L486: the term "extrametrical mycelium" has not been introduced in the text, so far.

L486-487 "P as a nutrient regulating fungal growth in boreal forest was not reported before": Please exactly define what fungal growth stands for in this case or reword for avoiding conflicts with earlier studies. For example, Aleida et al. 2019 (ref. in this manuscript) already wrote for the same forest: "Soil EMF communities responded more strongly to P than to N" which can be read as P is regulating fungal fungal growth. Not to mention the body of literature therein: "Ekblad et al. (1995) found that the production of extramatrical mycelium peaked under low P conditions. In a field study comparing Norway spruce (Picea albies) forests of varying P status, Rosenstock et al. (2016) observed greatly enhanced EMF biomass from ingrowth meshbags in the P limited forest [...] in P-limited forests, fungal biomass is enhanced by the presence of mineral P sources like apatite (Hagerberg et al., 2003; Berner et al., 2012; Rosenstock et al., 2016). Bahr et al. (2015) reported that apatite addition stimulated ingrowth of EMF in meshbags, especially in N-fertilized plots." (Almeida et al. 2019, https://doi.org/10.1016/j.funeco.2018.05.008). Why aren't those references seen as reports of P as nutrient which is regulating fungal growth?

**Reply:**

Those studies refer to EMF growth measured in the meshbags after a certain incubation time inside the soil. As it was mentioned in the introduction that standing biomass does not consider turnover of mycelium. In that sense their estimation of EMF growth (production) is different from our study.

Moreover, the "the P-limitation" reported in those studies is different from the P limitation reported here.

For example, in Rosenstock et al. (2016) the forest has P deficiencies due to soil parental material. Hagerberg et al. 2003 & Berner et al. 2012 measure EMF biomass in forest with different "P status" based on foliar P contents. And Ekblad et al. 1995 had a pot experiment where nutrient conditions were adjusted. In our forest P limitation is the result of N deposition and is confirmed by different means: Foliar chemistry, EMF communities and tree growth (in Almeida et al. 2019) and now by EMF growth. In that sense the current study is different and offers more evidence than the studies cited before.

However, this statement is confusing I agree and it contradicts those references in the introduction. Therefore, we have deleted that particular sentence: "*P* as a nutrient regulating fungal growth in boreal forest was not reported before". Its deletion does not affect the main message of the discussion.

L504: It would be worth mentioning that an independent second method measuring the decrease in belowground C allocation due to P is needed for verification in further studies.

Reply:

We have added this:

We propose that the decreased EMF production in the P-fertilized plots in our study is a result of a decrease in belowground C allocation due to reduced tree dependency on EMF for P foraging and acquisition. Fine root production and root tip colonization by EMF could be advisable as an independent second method to confirm that the decrease in EMF growth in the P-fertilized plots was an effect of reduced C allocation by the trees.

Technical corrections:

L21: "EMF and was" - missing word

Reply: Fixed.

L65: missing period

Reply: Fixed.

L118: EFM?

Reply: Fixed

L389-294 / Fig. 2: please label left and right panels (a, b). Remove the cluttering design of R-ggplot's standard output (grid lines, grey facet-boxes. Y-axis label: what is "per g" referring to? Quartz? Please choose a more exact way to label the y-axis like " $\mu$ g [Ergosterol] /  $\mu$ g [...]"

Reply: The labels have been fixed: Instead of Quartz it says Quartz-only to differentiate from the meshbags amended with apatite or urea. The grids were deleted and the Y legend now says: Ergosterol  $\mu g g^{-1}$

L398-403 / Fig. 3: Remove the legend and explain in the caption. Remove the cluttering design of R-ggplot's standard output (grid lines, grey facet-boxes. Y-axis label: what is "per g" referring to? Quartz? Please choose a more exact way to label the y-axis like " $\mu$ g [Ergosterol] /  $\mu$ g [...]"

Reply: The labels have been fixed: Instead of Quartz it says Quartz-only to differentiate from the meshbags amended with apatite or urea. The grids were deleted and the Y legend now says: Ergosterol  $\mu g g^{-1}$

L604 typo: not --> no

Reply: Fixed

**Reviewer 2:**

We want to thank the reviewer for his comments that improved the manuscript. In particular, the suggestion to add information about tipping points has made the discussion more interesting.

Comment:

The Discussion in particular is too long, it should be broken into sections e.g. hypotheses, limitations.

Reply:

The discussion was already broken in section related to the hypothesis. However, the link between hypothesis and the discussion sections was not that clear. Now we have related the discussion to the hypothesis in a better way.

The first section:

4.1 Effect of P fertilization on EMF biomass production and turnover

Relates to the first hypothesis:

Our first hypothesis was that P fertilization will decrease EMF biomass production in this P

limited forest as a result of the limitation being alleviated.

In the discussion:

In support of our first hypothesis, EMF biomass production declined in response to P fertilization in all meshbag amendments (Fig 4a).

The second section:

4.2 Effect of nutrient amendment on biomass production and turnover

Relates to the second and third hypothesis:

We expected that the nutrient patches will increase EMF biomass production depending on

fertilization. In particular: apatite amendment will increase EMF biomass production in the

control plots but not in P fertilized plots (second hypothesis); and urea amendment will

increase EMF biomass production in the P fertilized but not in the control plots (third

hypothesis).

In the discussion:

In support of our second hypothesis, apatite amendment increased EMF production in comparison with the pure quartz bags but only in the control plots.

From the two nutrient amendments, urea had the highest effect on EMF growth both in the control and P-fertilized plots partially confirming our third hypothesis.

The third section:

4.3 Seasonal effects on EMF growth

Relates to the fourth hypothesis (This is a newly added hypothesis):

Finally, since belowground C allocation follows the three phenological cycles (Endrulat et al., 2016), EMF production is likely to vary with season peaking in autumn (Hagerberg & Wallander, 2002;), we performed a more extensive incubation scheme and more frequent harvests of bags than in Ekblad et al., (2016). This allowed us to test not only effects of treatments (P fertilization) and of meshbag amendments, but also to estimate possible seasonal effects. Therefore, our fourth hypothesis was that EMF biomass production will be higher in autumn than in summer.

In the discussion:

In contrast with our fourth hypothesis, the EMF biomass production peaked in summer and decreased in autumn.

The limitations of the method were already discussed in the section concerning turnover:

These discrepancies relating EMF turnover rates between the current and previous studies might be caused by shortcomings on the sequential incubation method used for the model in this paper. This method relies on the premise that the sum of the biomass from meshbags incubated for short continuous periods should exceed the biomass from meshbags incubated from a long incubation time. However, in a number of cases the mycelial biomass from a long incubation period was greater than the sum of the consecutive shorter intervals.

We added more discussion concerning other limitations (lack of non-mycorrhizal controls). See my comments below.

**Comment:**

The paper overall needs to be more readable, easier to navigate for the reader. Redundancy/repetition should be reduced. There are also too many figures and tables, need to select the most important ones and move the rest to supplement.

**Reply:**

All the figures presented are important and relevant for the main message. I do not think that 6 figures is an exaggerated number. I would like to keep all the figures if the editor allows.

After all the changes suggested by the reviewers specially in the discussion the manuscript should read better.

**Comment:**

Unfortunately, no links are made to recent papers on nutritional thresholds or tipping points in EM for European (incl. spruce Swedish) forests.

Answer:

We have added information about the tipping points. See my comment below.

**Comment:**

There are no links made to recent studies of P limitation (i.e. beech in Germany).

**Answer:**

The main goal of the paper was to discuss how P limitation affected EMF growth; it was not to discuss P limitation itself. The limitation itself was the main story of Almeida et al 2019 where this was further discussed. For that reason, P limitation of the trees is briefly mentioned in the current manuscript.

However, the references are not that updated I must admit.

We have added new references including studies such as: Pinus sylvestris forests in Germany (*Prietzel et al.* 2020), different tree species (deciduous and evergreen) across Europe (Du et al., 2021), Fagus sylvatica across Europe (*Talkner et al.* 2015)

Prietzel, J., Falk, W., Reger, B., Uhl, E., Pretzsch, H., & Zimmermann, L. (2020). Half a century of Scots pine forest ecosystem monitoring reveals long-term effects of atmospheric deposition and climate change. Global Change Biology, 26(10), 5796-5815.

Du, E., van Doorn, M., & de Vries, W. (2021). Spatially divergent trends of nitrogen versus phosphorus limitation across European forests. Science of the Total Environment, 771, 145391. Talkner, U., Meiwes, K. J., Potočić, N., Seletković, I., Cools, N., De Vos, B., & Rautio, P. (2015). Phosphorus nutrition of beech (Fagus sylvatica L.) is decreasing in Europe. Annals of forest science, 72(7), 919-928.

In those references more evidence of P limitation (as a result of N deposition) in temperate forest is offered.

We have changed the phrasing of this in the introduction and discussion since these new references further support P-limitation in northern ecosystems:

In the introduction before we said:

However, it has been suggested that anthropogenic N deposition can potentially change the forests nutrient requirements and push the system toward phosphorus (P) limitation (Tarvainen et al., 2016; Du & Fang, 2014; Akselsson et al., 2010; Vitousek et al., 2010)

Now we say:

However, anthropogenic N deposition can potentially change the forests nutrient requirements and push the system toward phosphorus (P) limitation (Tarvainen et al., 2016; Du & Fang, 2014; Akselsson et al., 2010; Vitousek et al., 2010 ; Talkner et al. 2015; Prietzel et al. 2020 ; Du et al., 2021)

In the discussion we have added:

Therefore, the effects of P fertilization alone on boreal forests have not been widely tested **despite evidence that** the steep increase in anthropogenic C and N inputs can lead to unbalanced nutrition and push forested ecosystems towards P limitation (Jonard et al., 2015; Peñuelas et al., 2013; **Talkner et al. 2015; Prietzel et al. 2020 ; Du et al., 2021).**

Comment:

There is no mention of mycorrhizal fungi supressing decomposers (Gadgil effect).

Answer:

The main objective of the manuscript was to measure EMF biomass and relate it to P limitation of the trees. The changes in belowground C allocation that we discuss here could

potentially affect the ecological relations between EMF and saprotrophs but that would be an experiment and discussion on its own.

EMF might have competitive advantage over saprotrophs in the bags since the bags contain less C resources than the bulk soil but that was not the main objective of the experiment.

Comment:

There is frequent mention of tree C allocation, as if this was being measured directly, but this is not the case. It is indirect and incomplete (via fungal growth and without respiration); so this needs to be qualified consistently throughout, e.g. "potential tree C allocation".

Reply:

Potential C allocation has been added when necessary. Please not that in some cases we say:

We propose that the decreased EMF production in the P-fertilized plots in our study is a result of a decrease in belowground C allocation...

**OR:**

*When P limitation is alleviated by fertilization however, there is probably a decrease in C allocation to the root*

In these cases adding the word potential is not necessary since we are suggesting it as a possibility.

Comment:

It is often unclear if apatite amendment is the same as P fertilization, which it is not; both are used interchangeably, which is confusing.

**Reply:**

We have modified the parts that are confusing. Now all the parts that refer to superphosphate are called P fertilization and all the parts that refer to apatite are called apatite amendment or apatite-amended bags

Comment:

The abstract needs to mention how P fertilization was done, e.g superphosphate twice.

Reply:

We added:

*Here we estimated EMF production and turnover in response to P fertilization (applied as superphosphate)*

Comment:

24 - high P status is vague - of soil, fungus, tree?

Reply:

Agree, is vague. We have changed to:

*P* fertilization had a negative effect on EMF production in all the meshbag amendments suggesting a reduced belowground C allocation to the EMF when P limitation is alleviated.

Comment:

What is the standard/threshold used? What is the evidence for P limitation?

Reply:

These results and the discussion about tree limitation in this forest was already discussed in Almeida et al. (2019).

In that paper the tree responded significatively to P fertilization while no effect was detected on tree growth when the trees were fertilized with nitrogen which is an indication also that the trees were P limited. EMF communities were also significantly affected by P fertilization confirming the importance of this nutrient in this forest. In the current paper we measure and discuss EMF growth in the same forest plots as Almeida et al. (2019). The main focus of the current paper is to discuss the effect on EMF growth in this forest where the evidence for P limitation was already presented in our previous paper.

However, the tipping points topic is very interesting therefore I have added information about the N:P ratios measured in Almeida et al. (2019). See my comments below

**Comment:**

**72. Quantifying biomass and growth?**

**Reply:**

Growth. The point is of this paragraph is to discuss growth (increase in biomass through time). Measuring growth rates of free-living fungi can be done in the lab but EMF need the host to grow.

**Comment:**

78. Is there evidence for nearly zero non-ECM growth into bags?

**Reply:**

No there is no evidence for zero non-ECM growth. However, the meshbag methods is well known for selecting EMF from other fungal guilds. Thus, it is likely that **most of the growth** response registered is of EMF origin.

We did acknowledge that in the discussion:

The main assumption that the ergosterol in this experiment comes mostly from EMF relies on previous evidence that the meshbag system favors the growth of EMF over non-mycorrhizal fungi (Almeida et al., 2018; Rosenstock et al., 2016; Berner et al., 2012; Wallander et al. 2010; Hedh et al. 2008).

However, as pointed out by the other reviewer, we do not have non-mycorrhizal controls (fungal biomass measured from ingrowth bags collected in a trenched-root-free area of soil) or DNA data to confirm how much EMF growth there was.

Therefore, we reorganize that last part of the discussion and acknowledge this:

**4.4 Potential non-mycorrhizal growth in the meshbags**

It could be also possible that non-mycorrhizal fungi contributed to the fungal growth detected

in the current study. The main assumption that the ergosterol in this experiment comes mostly

from EMF relies on previous evidence that the meshbag system favors the growth of EMF

over non-mycorrhizal fungi (Almeida et al., 2018; Rosenstock et al., 2016; Berner et al.,

2012; Wallander et al. 2010; Hedh et al. 2008; Wallander et al., 2001). However, it has been shown that the shorter the time period a meshbag remains underground the higher the proportion of non-mycorrhizal fungi inside the bags (as measured by the proportion of nonmycorrhizal DNA in Hagenbo et al., 2018).

Thus, non-mycorrhizal fungi growth could partially explain the seasonal effect detected as this fungal guild has been reported to respond positively to temperature (Pietikäinen et al., 2005). Unfortunately, the current study lacks non-mycorrhizal biomass controls (ie: fungal biomass from ingrowth bags collected in a trenched root-free area) that can be used to estimate the contribution of non-mycorrhizal fungi. Therefore, we cannot rule out the possibility that part of the ergosterol measured in the bags came from non-mycorrhizal fungi. Even so, the significant negative effect of P fertilization on all the meshbag types suggests that the decrease in fungal growth might be related to potential reduction in C allocation by the trees as discussed earlier. Moreover, the effects of the P fertilization and meshbag amendment on fungal growth were higher early in the season which might imply that the seasonal effect seen in the current study is explained mostly by EMF.

It must be noted nevertheless that a potential reduction in belowground C allocation could decrease root activity and possibly root exudates which might reduce labile sugars in the soils affecting saprotrophic fungi as well. Further studies are necessary to evaluate the effect of P limitation on root dynamics and other members of soil microbial communities.

Comment:

91-3. Is there any evidence supporting these assumptions?

Reply:

Biomass and necromass losses are both just organic matter decomposition, so we would expect them to follow the same kinetics than any other organic matter (first order, so linear first derivative and exponential integrated over time).

EMF growth is assumed to be zero order (so first derivative = constant number, always the same rate) because we assume growth is limited by transport of nutrients from outside, that is the kinetic limitation.

With bacterial (or single cell organisms) growth, the limit is only the substrate, so we use a first order kinetic. The more substrate, the more individuals will grow and the more the rate: dC/dt=C\*k, the derivative depends on the growth rate of each cell and the substrate. The substrate varies with time and this give a variable rate and an exponential growth. With hyphae the derivative is assumed as dC/dt=k, because k is in this case the rate of transport possible through the mesh.

Either of these assumptions (first order or zero order kinetic for hyphae) has its own amount of inexactness, being both approximations, but the latter seems more correct and a more realistic model.

Comment:

157. Total experimental site area?

Reply:

We added:

**The total experimental area comprised 2.1 ha1. The experiment consisted of 6 plots.....**

Comment:

163. Is this above published thresholds for EM?

**Reply:**

These values exceed the N critical loads in which negative changes in the function and composition of an ecosystem are expected (Kuylenstierna *et al.*, 1998; Pardo *et al.*, 2011; Pihl Karlsson *et al.*, 2017).

- Kuylenstierna, J. C. I., Hicks, W. K., Cinderby, S., & Cambridge, H. (1998). Critical loads for nitrogen deposition and their exceedance at European scale. In *Nitrogen, the Confer-Ns* (pp. 591-598).
- Pardo, L. H., Fenn, M. E., Goodale, C. L., Geiser, L. H., Driscoll, C. T., Allen, E. B., et al. (2011). Effects of nitrogen deposition and empirical nitrogen critical loads for ecoregions of the United States. *Ecological Applications*, 21(8), 3049-3082.

Pihl Karlsson, G., Akselsson, C., Hellsten, S., & Karlsson, P. E. (2017). Krondroppsnätet i södra Sverige: Övervakning av luftföroreningar och dess effekter i skogsmiljön. Resultat till och med september 2016. Moreover, in our previous paper on this forest plots the tree responded significatively to P fertilization while no effect was detected on tree growth when the trees were fertilized with nitrogen which is an indication that the trees were P limited. EMF communities were more different in the P fertilized plots than in N fertilized plots (both soil and meshbags) confirming that in this forest P has a more important role than N in EMF communities.

Comment:

166. Why 200kg?

Availability of P can be very low in soil. To make sure we get enough available P for the trees to get a growth effect if P is limiting, an excess of P was added.

Comment:

171. Why 50um?

Reply:

It is the size used to exclude fine roots but allow mycelium hyphae colonization. This information is given in the original paper that describes meshbags (Wallander et al., 2001) That reference is provided in the methods.

Comment:

172. Acid-washed quartz?

Reply:

Yes. We modified it in the text.

Comment:

173. Why 2%, 50-250nm, 0.5%?

Reply:

There was a mistake in the apatite grain size units. It should be um instead of nm (we did not crush apatite to the nanoscale!). And it should be 1.5 instead of 2%.

The amount of apatite we used is similar to amount the Rosenstock et al. (2016). We chose that amount to give enough mineral to the fungi to sustain growth for the whole duration of the experiment.

We gave more apatite than use since apatite is a more recalcitrant source and has less % of P (18) in comparison with the % of N (42%) that the methylene use has.

The apatite size was chosen based on other studies showing that the finer the grain size the highest the EMF respiration colonizing the mineral (Leake et al.,2008). 50 um is small enough for the material not to pass through the bags' mesh. 250 mg was the smaller sieve size we had to give a higher threshold for the crushed mineral.

Comment:

181. Why 30, 60, 90, 120 or 150d?

Reply:

150 days was the duration of our experiment which contained the productive period for EMF growth.

28 days is discussed as the mean residence time of the biomass in Ekblad et al. 2016. We rounded up to 30.

The rest is because we wanted to have as many overlapping incubation periods as possible for the model to have enough data points. A bag incubated for 150 days overlaps with five bags incubated for 30 days. A bag incubated for 120 days overlaps with four bags incubated for 30 days and so on.

Comment:

215-6. Would there be significant disturbance effects?

Reply:

Yes, it can be a disturbance and it is specified as a weakness of the experimental design and acknowledged in the discussion:

This method relies on the premise that the sum of the biomass from meshbags incubated for short continuous periods should exceed the biomass from meshbags incubated from a long incubation time. However, in a number of cases the mycelial biomass from a long incubation period was greater than the sum of the consecutive shorter intervals. This could be caused by a delay or a lag phase in EMF colonization inside the bags. It is possible that when a meshbag was collected and the same hole was used to replace a new bag (Fig 2) there was a lag phase before the hyphae could colonize the newly placed meshbag (Wallander et al., 2013). Thus, those data points could have created noise in the data making the turnover estimates less robust.

Comment:

219. Up to how long?

Reply:

Up to 10 hours in summer and 8 in winter that is roughly the amount of day light hours when the sampling was done.

Comment:

221. Why not chitin or nucleic acids?

Reply:

We used ergosterol as an estimation of living fungal biomass. Ergosterol has been used to estimate living biomass in many studies.

Chitin is more recalcitrant source and was used to enable estimation of the necromass in Ekblad et al. (2016). It would have been ideal to use both, but we did not have developed that protocol in our lab.

Nucleic acids would not be a good option. To identify fungal DNA amplicon sequencing would be necessary. This technique is based on relative abundances and is semiquantitative. Therefore, it won't be useful to quantify total biomass as we did in this paper.

Comment:

302. Which relative (related?) publications?

Reply:

We meant relative information of papers that have used similar modeling like Ekblad et al. (2016) or Hagenbo's. However, that statement is confusing and do not add much so I remove it.

Comment:

367. Were there normal average temp and ppt at the site in 2015 summer/autumn?

Reply:

According to the climate data the average temp. that year was not different from other years. Peaked in July and decrease in autumn. The July and September average temps. were very similar among the years 2013,14,15,16.

For the precipitation 2014 had more precipitation in July (and in general) than 2013, 2015 and 2016. There were some differences in the precipitation patterns among years but nothing very clear to help us understand the seasonal effect seen in our data. Moreover, sampling meshbags more years than 2015 would have been necessary to understand the potential effects of climatic variation between years on EMF growth.

Comment:

Figure 2. Why show this as 1 vs 2 months? Redundant?

Reply:

It is not redundant. The two-months in this case **do not** come from the sum of 2 one-month bags. They are bags that have been incubated for two months. So both incubation times (bags incubated for one month and bags incubated for two months) confirm the high biomass early in the season.

Comment:

Figure 3. Could saprobes use carbon in urea?

Reply:

This is a very good observation. Yes, there it is carbon in the urea meshbags. However, the methylene urea contains only 2 carbons. Only few organisms (methylotrophs bacteria) can use it as the sole source of C and N. It is unlike than fungi could have used it as both N and C sources.

Comment:

Discussion: Any practical recommendations for future studies? Tradeoffs in replication/design/power?

Reply:

We have added a recommendation for studies that directly test below ground C allocation:

We propose that the decreased EMF production in the P-fertilized plots in our study is a result of a decrease in belowground C allocation due to reduced tree dependency on EMF for P foraging and acquisition. Fine root production and root tip colonization by EMF could be advisable as an independent second method to confirm that the decrease in EMF growth in the P-fertilized plots was an effect of reduced C allocation by the trees.

Comment:

479. shown indirectly in the current results?

Reply:

Changed.

Comment:

480. P means superphosphate and/or apatite?

Reply:

Superphosphate. We changed the text:

In the present study, *P* fertilization had a negative effect on the EMF standing biomass in most of the incubation periods

Comment:

505. and/or shifted EM community?

Reply:

Yes, besides decreasing total EMF biomass, belowground C allocation can potentially alter EMF community composition if the tipping points for nutritional changes have been reached.

We have added:

A decrease in below ground C allocation is also expected to alter EMF community composition selecting for C efficient species when the ecosystem has crossed the nutritional tipping point thresholds (Suz et al., 2021).

Comment:

511. P-limitation - were trees tested, e.g. foliar chemistry?

Reply:

They were tested in Almeida et al. (2019) and the values were found to be under the threshold for P limitation of Norway spruce. These results and the discussion about tree limitation in this forest was already discussed in Almeida et al. (2019) and there was not the objective of the current manuscript to repeat those results.

However, the tipping points topic is very interesting therefore I have added information about the N:P ratios measured in Almeida et al. (2019) and related them to the results of the current paper:

Indeed, in the study performed by Almeida et al. (2019) in the same experimental plots as the current experiment, it was reported that P fertilization enhanced tree growth. Moreover, the authors reported that the foliar N:P ratios measured in the unfertilized control plots corresponded to suggested tipping points where the ecosystem shifts towards P limitation (see Suz et al., 2021 & van der Linde et al., 2018).

Comment:

523-524 - observed or assumed?

**Reply:**

That part has changed along with all that paragraph. See my comments below.

Comment:

528. smooth mantle, not mycelium

Reply:

Changed.

**Comment:**

514-538. This paragraph could be shorter (less speculative) and more sophisticated/precise. Why talk about Tylospora when no fungal community data is presented? I am hoping there won't be a follow up paper where fungal community data is presented separately, if it is available, it should be presented here.

**Reply:**

It is true that no fungal communities were analyzed in the current experiment. However, in Almeida et al (2019) we did analyze EMF soil community data in the **very same plots** of the present study and **very close in time as the data collection of the current paper (2014)**.

Ideally, we should have community data from the meshbags analyzed in the current study now but since we do not, the EMF soil community data shown in Almeida et al (2019) should represent what is happening in the soils of our experimental plots.

There will not be a follow up paper with communities because this is the follow up paper that comes after we analyzed the EMF communities

Moreover, the tipping points, suggested to be discussed, clearly state that under certain thresholds EMF communities should change, which indeed happen in our experimental plots. Therefore, if tipping points are to be discussed we must mention the community data in the discussion of the current paper. In Almeida et al (2019) *Tylospora asterophora* responded to P fertilization (both P fertilization alone and in combination with nitrogen) in the soils and in the meshbags. Therefore, it should be mentioned in this paragraph.

We have changed this section of the discussion, reduced some information about Tylospora but added new one that is important and finally added this part of the discussion right after we discuss the tipping points: A decrease in below ground C allocation is also expected to alter EMF community composition selecting for C efficient species when the ecosystem has crossed the nutritional tipping point thresholds (Suz et al., 2021). Indeed, in the soil EMF survey performed in the same experimental plots as the present study, Almeida et al. (2019) reported that the relative abundance of Tylospora asterophora was significantly increased after P fertilization. This species has been reported to extensively occupy ingrowth meshbags while colonizing relatively low amount of tree root tips which might suggest either a high C efficiency or lower turnover rates (Jörgenssen, 2021). The lack of difference in turnover rates between fertilized and unfertilized plots in the present study might suggest the earlier.

**Comment:**

601. But not measured, is this this a hypothesis?

**Reply:**

True it is not measured but the below ground C allocation was part of the theoretical background that lead to the first hypothesis:

It was suggested that the decrease in EMF growth was regulated by changes in carbon allocation as a result of an increase in soil fertility. These results are in line with evidence indicating that the relative amount of carbon allocated to EMF is sensitive to plant nutrient status and soil fertility (Gill & Finzi 2016).....

......The effect of the transition from N to P limitation on the below ground C allocation and EMF growth has not been studied in natural soils, but P deficiency is expected to increase EFM biomass to improve P foraging and uptake (Rosenstock et al., 2016; Ekblad et al. 1995; Wallander & Nylund 1992).

**Comment:**

616-617 - explain in methods section.

Comment:

700 - P - meaning superphosphate and/or apatite?

Reply:

Changed:

In conclusion, EMF production was strongly reduced when the **P** fertilizer was added to the forest.

---

## Author Response (AR2)

Reviewer 1:

I remain with a minor comment:
Though I understand that figure layouts are a design choice, I am a bit at odds with the figure ratios in Fig. 2 and 3, which now both show the same plot types, y-axes and the same range of values. However, concerning the use of space, the y-axes are very squeezed in Fig. 2 while they are very strechted in Fig. 3 (around five times higher), for which do not see a reason.
With more similar y-axis heights, both figures could even be shown as one combined figure consisting of a) and b). - As additional benefit, this would be in line with reviewer 2's recommendation to reduce figures.

That is a good observation, however even if the Y-axis of both figures represent the same range of values the data set shown in them is not the same.

Figure 2 is a composite figure of 12 different panels each with its own y axis. The axis seems contracted because 6 of those panels are fitted vertically in one figure.

Figure 3 is only one panel and for that reason the Y axis seems expanded. To make figure 3 axis more similar to figure 2 then we would need to reduce figure 3 size to a point it won't look good.. Or figure 2 would need to be enhanced so much that it would not fit a page.

It would be possible to decrease the number of panels in figure 2 of course. For example, including all 3 meshbag amendments in one panel instead of 3 but this will make the panel more crowded and difficult to distinguish what is important in the figure: the differences between months. For that reason, this dataset configuration was decided.

Therefore, I must respectfully insist on keeping the current figure configuration.

However, I must admit that in the last manuscript version the figure 2 was not large enough and the difference between axis was more evident. In the newest version I have enlarged figure 2 and it does not look that weird anymore.

Reviewer 2:

EMF might have competitive advantage over saprotrophs in the bags since the bags contain less C resources than the bulk soil but that was not the main objective of the experiment

- So, mention Gadgil Effect briefly.

Reply: The Gadgil effect is defined as the suppression of the **soil organic matter decomposition rates** caused by EMF outcompeting saprotrophic fungi (Fernandez and

Kennedy, 2015 in: Revisiting the 'Gadgil effect': do interguild fungal interactions control carbon cycling in forest soils?).

We did not measure decomposition rates in the meshbags. As mentioned before, there could have been competitive advantage of EMF over saprotrophs inside the bags due to the lack of organic matter in the bags. However, that is not related to **organic matter decomposition suppression by EMF.** Therefore, there is no place to mention the Gadgil effect unfortunately.

Growth. The point is of this paragraph is to discuss growth (increase in biomass through time). Measuring growth rates of free-living fungi can be done in the lab but EMF need the host to grow.

- Clarify in text, other readers may be confused too.

Reply: That is precisely mentioned in the paragraph:

*Growth rates of free-living fungi from natural soils has been studied in laboratory by measuring labeled acetate incorporated in the fungal membrane component ergosterol Quantifying growth (production) of EMF natural communities on the other hand is more complicated since EMF are dependent on plant roots (Smith and Read, 2008) and such measurements must be performed when the fungi is living in symbiosis.*

Either of these assumptions (first order or zero order kinetic for hyphae) has its own amount of inexactness, being both approximations, but the latter seems more correct and a more realistic model.

- Explain this in the text.

Reply: Added explanation in the material and methods where the details of the model are explained.

These values exceed the N critical loads in which negative changes in the function and composition of an ecosystem are expected (Kuylenstierna et al., 1998; Pardo et al., 2011; Pihl Karlsson et al., 2017).

- Write this in the ms.

Reply: Done.

Availability of P can be very low in soil. To make sure we get enough available P for the trees to get a growth effect if P is limiting, an excess of P was added.

- Clarify in text.

Reply: Done in the material and methods.

It is the size used to exclude fine roots but allow mycelium hyphae colonization. This information is given in the original paper that describes meshbags (Wallander et al., 2001) That reference is provided in the methods.

- Explain this in text or a methods supplement, so readers don't have to look up a 2001 paper.

Reply: Explained in the methods.

The amount of apatite we used is similar to amount the Rosenstock et al. (2016). We chose that amount to give enough mineral to the fungi to sustain growth for the whole duration of the experiment. We gave more apatite than urea since apatite is a more recalcitrant source and has less % of P (18) in comparison with the % of N (42%) that the methylene urea has. The apatite size was chosen based on other studies showing that the finer the grain size the highest the EMF respiration colonizing the mineral (Leake et al., 2008). 50 um is small enough for the material not to pass through the bags' mesh. 250 mg was the smaller sieve size we had to give a higher threshold for the crushed mineral.

- Clarify in text or a methods supplement.

Reply: Added in the text.

150 days was the duration of our experiment which contained the productive period for EMF growth. 28 days is discussed as the mean residence time of the biomass in Ekblad et al. 2016. We rounded up to 30. The rest is because we wanted to have as many overlapping incubation periods as possible for the model to have enough data points. A bag incubated for 150 days overlaps with five bags incubated for 30 days. A bag incubated for 120 days overlaps with four bags incubated for 30 days and so on.

- Clarify in text or a methods supplement.

Reply: Added a couple of sentences in the materials and methods where I mention the experimental setup.

Yes, it can be a disturbance and it is specified as a weakness of the experimental design and acknowledged in the discussion.

- Be explicit, write 'disturbance' in the ms.

Reply: Done. Added that there could be a disturbance of the mycelial connections when collecting the bags.

Up to 10 hours in summer and 8 in winter that is roughly the amount of day light hours when the sampling was done.

- Clarify in text or a methods supplement.

Reply: Added in the materials and methods.

We used ergosterol as an estimation of living fungal biomass. Ergosterol has been used to estimate living biomass in many studies. Chitin is more recalcitrant source and was used to enable estimation of the necromass in Ekblad et al. (2016). It would have been ideal to use both, but we did not have developed that protocol in our lab. Nucleic acids would not be a good option. To identify fungal DNA amplicon sequencing would be necessary. This technique is based on relative abundances and is semiquantitative. Therefore, it won't be useful to quantify total biomass as we did in this paper.

- Clarify in text or a methods supplement.

Reply: Added a couple of sentences in the materials and methods where I mention that ergosterol was used.

According to the climate data the average temp. that year was not different from other years. Peaked in July and decrease in autumn. The July and September average temps. were very similar

among the years 2013,14,15,16. For the precipitation 2014 had more precipitation in July (and in general) than 2013, 2015 and 2016. There were some differences in the precipitation patterns among years but nothing very clear to help us understand the seasonal effect seen in our data. Moreover, sampling meshbags more years than 2015 would have been necessary to understand the potential effects of climatic variation between years on EMF growth.

- Mention in text or a results supplement.

Reply:

Added in the discussion when we discuss the seasonal patterns observed.

It is not redundant. The two-months in this case do not come from the sum of 2 one-month bags. They are bags that have been incubated for two months. So both incubation times (bags incubated for one month and bags incubated for two months) confirm the high biomass early in the season.

- Clarify in the text or figure legend.

Reply: Now the figure legend is more specific:

*Figure 2: Standing EMF biomass in the **meshbags that have been in the soil for a period of 2 months a) and a period of 1 month  b**). The error bars represent the standard error of the mean (n=3).Lowercase letters represents statistically significant (P<0.05) differences between the incubation periods according to Dunn´s test.*

This is a very good observation. Yes, there it is carbon in the urea meshbags. However, the methylene urea contains only 2 carbons. Only few organisms (methylotrophs bacteria) can use it as the sole source of C and N. It is unlike than fungi could have used it as both N and C sources.

- Mention this in text or a results supplement.

Reply:

UPTDATE: Recently I found out that there are methylotrophic yeasts and they can be present in soils. So it is possible that the use of methylene urea as a carbon source is not restricted to bacteria.

Therefore, in the discussion section where I already acknowledge the potential presence of non-mycorrhizal fungi in the meshbags, I have added a sentence about these yeasts:

*Therefore, we cannot rule out the possibility that part of the ergosterol measured in the bags came from non-mycorrhizal fungi (ie: **methylotrophic yeasts in the urea-amended bags that can use methylene urea as both C and N sources**). Even so, the significant negative effect of P fertilization on all the meshbag types suggests…..*

line 112. 'it has been suggested that anthropogenic N deposition can potentially change the forests nutrient requirements and push the system toward phosphorus (P) limitation' - but see Scientific Reports 7, 7856 (2017), Science 376, eabh3767 (2022).

Reply: Good point. There seems to be some evidence about N decreasing in soils in north America (Reports 7, 7856, 2017) and about N depositions peaks decreasing and stabilizing in Europe (Science 376, eabh3767, 2022). However, there are also some other studies

suggesting that northern forests ecosystems (thought to be N-limited) are showing signs of P limitation. Therefore, I have made it more specific in this part of the intro:

*However, **there is some evidence suggesting** that anthropogenic N deposition can potentially change the forests nutrient requirements and push the system toward phosphorus (P) limitation (Almeida et al., 2019; Jonard et al., 2015; **Talkner et al. 2015; Prietzel et al. 2020 ; Du et al ., 2021**)*